# Prune, Interpret, Evaluate (PIE 🥧): A Cross-Layer Transcoder-Native Framework for Efficient Circuit Discovery via Feature Attribution

## Abstract

Existing feature-interpretation pipelines typically operate on uniformly sampled units, but only a small fraction of cross-layer transcoder (CLT) features matter for a target behavior, with the rest resulting in expensive feature explaining and evaluating costs. We introduce the first CLT-native end-to-end framework, **PIE**, connecting Pruning, automatic Interpretation, and interpretation Evaluation, enabling systematic measurement of behavioral fidelity and downstream interpretability under pruning. To achieve this, we propose Feature Attribution Patching (FAP), a patch-grounded attribution method that scores CLT features by aggregating gradient-weighted write contributions, and FAP-synergy, a synergy-aware reranking procedure. We evaluate pruning using KL-divergence behavior retention and assess interpretation quality with FADE-style metrics. On IOI with CLTs for Llama-3.2-1B and Gemma-2-2B, pruning to $K{=}100$ features matches the KL fidelity that a strong random baseline requires $\approx 4$k active features to achieve ($\approx 40\times$ compression), enabling $\approx 40\times$ fewer interpretation/evaluation calls while substantially reducing low-quality features. FAP-Synergy further improves fidelity at fixed budget and yields better interpretability-fidelity trade-offs. Our code is available at https://anonymous.4open. science/r/ICML-submission-D0AC.

## 1. Introduction

Mechanistic interpretability aims to interpret model behavior by identifying internal components and causal pathways that mediate computation from inputs to outputs (Olah, 2022). This program has gained urgency as large language models (LLMs) are deployed in increasingly high-stakes settings where transparency, auditing, and targeted intervention matter (e.g., safety and biomedical decision support) (Templeton et al., 2024; Yang et al., 2022; Band et al., 2023; Chen et al., 2025). Recent progress has moved from neuron-level anecdotes toward scalable *feature- and circuit-level* accounts, enabled by sparse feature bases such as sparse autoencoders (SAEs) and replacement-model approaches such as cross-layer transcoders (CLTs) (Bricken et al., 2023; Ameisen et al., 2025; Conmy et al., 2023; Syed et al., 2024).

However, despite rapid advances in circuit discovery, a central bottleneck remains: *which internal units should we spend interpretation budget on?* Modern feature dictionaries can contain hundreds of thousands to millions of features, while downstream auto-interpretation and evaluation pipelines are expensive per feature (Bills et al., 2023; Paulo et al., 2024; Puri et al., 2025; Boggust et al., 2025). In practice, an end-to-end interpretability workflow must allocate compute carefully: explaining *everything* is infeasible, and explaining uninformative features is wasteful.

**Motif 1: pruning is necessary for scalable auto-interpretation, and CLTs demand a CLT-native framework.** Existing automatic interpretation pipelines largely focus on SAE features or neuron-like units and typically evaluate uniformly sampled features, which leads to substantial redundancy: many features are weakly causal, noisy, or irrelevant to the downstream behavior of interest (Bills et al., 2023; Paulo et al., 2024; Puri et al., 2025; Boggust et al., 2025). This motivates a simple principle: *not all features are worth explaining; prune first, interpret later.*

At the same time, CLT replacement models represent circuits using cross-layer feature writes that are not naturally captured by SAE-only pipelines (Ameisen et al., 2025; Lindsey et al., 2025). We therefore argue that scalable mechanistic interpretability requires the first CLT-native auto-interpretation framework that explicitly connects: pruning → automatic interpretation → interpretation evaluation. This paper establishes such a framework, enabling end-to-end measurement of how pruning decisions shape both be-

[1]Anonymous Institution, Anonymous City, Anonymous Region, Anonymous Country. Correspondence to: Anonymous Author <anon.email@domain.com>.

Preliminary work. Under review by the International Conference on Machine Learning (ICML). Do not distribute.

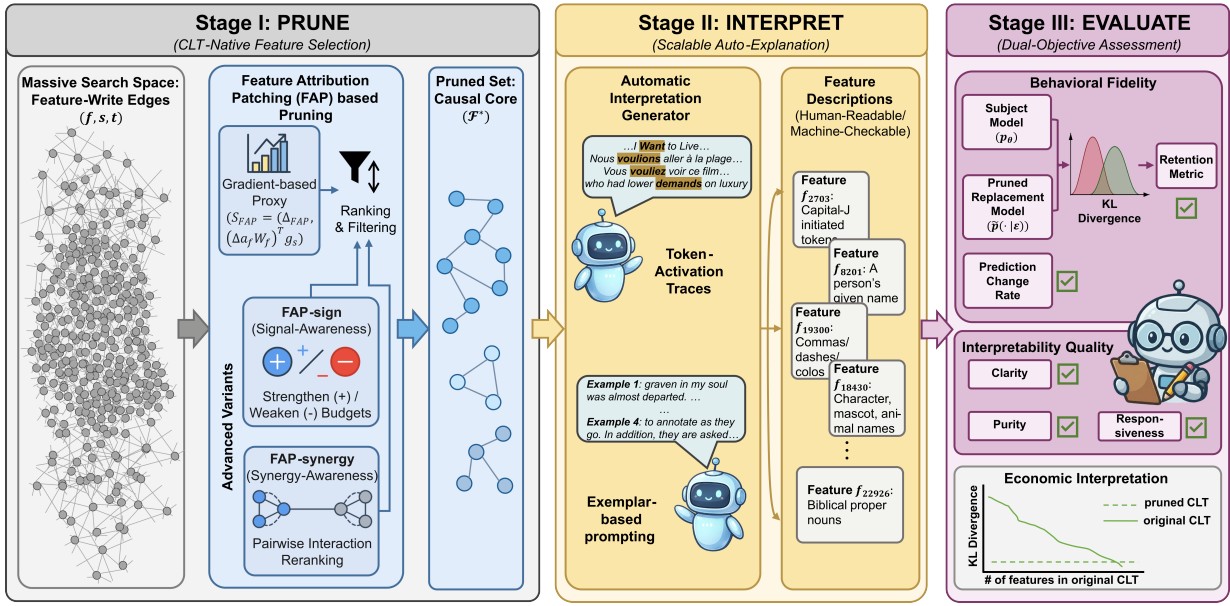

*Figure 1.* The PIE Framework: A CLT-Native End-to-End Pipeline. We propose a three-stage framework to enable scalable interpretability: **Stage I (Prune)** filters the massive search space of CLT feature-write edges into a sparse "Causal Core" using Feature Attribution Patching (FAP) and its synergy-aware variant (FAP-Synergy), which reranks boundary features based on pairwise interactions. **Stage II (Interpret)** generates natural language descriptions only for the retained features using exemplar-based prompting, drastically reducing interpretation costs. **Stage III (Evaluate)** performs a dual-objective assessment: quantifying behavioral fidelity (KL Divergence, PCR) and measuring interpretation quality via automated metrics (Clarity, Purity, Responsiveness).

havior fidelity and downstream explainability metrics.

**Motif2: mainstream pruning is insufficient because it ignores synergy and sign.** A second bottleneck lies in how pruning is performed. Most mainstream pruning methods are edge-centric and/or assume approximately additive importance, selecting high-magnitude contributors and discarding the rest (Syed et al., 2024; Hanna et al., 2024). However, CLT circuits can exhibit two properties that violate these assumptions: (i) *non-additive interactions*, where components that appear weak in isolation matter jointly ("synergy"); and (ii) *signed influence*, where both strengthening and weakening contributions are necessary for calibrated behavior. Such effects are well-known in causal tracing / patching settings, and we empirically verify their presence in CLT feature space in our ablations (Appendix A and Sec. 4).

Ignoring these effects can systematically under-estimate important components and produce circuits that are behaviorally degraded or interpretability-unstable.

**Our approach: Feature Attribution Patching (FAP) and its extensions.** To address both motifs, we introduce a CLT-native pruning family that operates directly on *features* rather than module-level edges. Our base method, Feature Attribution Patching (FAP), scores CLT feature occurrences using a fast, patch-grounded first-order estimate

based on cross-layer write differences dotted with cached downstream gradients (an attribution patching analogue in CLT feature space). We then propose FAP-sign, a *signal-aware* variant that preserves both strengthening and weakening features by allocating budget separately to positive and negative contributions. We also study a sign-aware ablation, FAP-Signal, which enforces a minimum allocation to both positive- and negative-scoring features; details and results are in Appendix D.

This paper makes four contributions:

- CLT-native prune-interpret-eval framework. We present the first end-to-end framework connecting CLT-native pruning to automatic interpretation and interpretation evaluation, explicitly motivated by the redundancy of SAE-style interpretation pipelines when applied at scale.

- Feature Attribution Patching (FAP). We propose an efficient feature-level attribution patching method for CLT circuits that scores sparse cross-layer feature writes with a single gradient pass, enabling large-scale pruning.

- **Synergy-aware pruning.** We introduce **FAP-Synergy**, a boundary interaction reranking method that improves fidelity at fixed budget. We additionally evaluate a sign-aware ablation, **FAP-Signal**, in Appendix D.

- Comprehensive evaluation protocol. We evaluate pruning using KL-divergence behavior retention and assess interpretability impact via interpretation metrics including clarity, purity, and responsiveness (Puri et al., 2025; Paulo et al., 2024; Boggust et al., 2025), enabling systematic quantification of how pruning affects explainability.

## 2. Related Work

**Attribution patching and edge pruning.** Causal circuit discovery methods often formalize the search space as a computation graph and aim to recover a sparse subgraph sufficient for reproducing a behavior under intervention. Early work such as ACDC performs iterative edge search using patching-based causal tests (Conmy et al., 2023), but this can require many forward passes and becomes expensive at scale. Attribution Patching-style methods accelerate this process by computing proxy importance scores using gradients and validating them with patching, substantially reducing search cost while retaining causal grounding (Syed et al., 2024). Our work inherits the patching-first philosophy but changes the unit of analysis: rather than pruning *module-to-module edges*, we prune *CLT feature writes*, which are the natural causal objects in a cross-layer replacement model.

**Feature-based interpretability and CLT-native circuit tracing.** A recurring theme in mechanistic interpretability is that individual neurons are often polysemantic, motivating learned sparse feature bases as more stable semantic units (Bricken et al., 2023). Toy models of superposition and related training phenomena provide a conceptual basis for why feature directions, rather than single neurons, may serve as the correct minimal units for interpretation (Elhage et al., 2022; Henighan et al., 2023; Power et al., 2022). Replacement-model approaches further enable causal validation by allowing controlled interventions on feature activity and information flow. In particular, CLT-based circuit tracing represents computation as sparse cross-layer feature writes and supports end-to-end causal experiments in which feature contributions are patched, ablated, or amplified (Ameisen et al., 2025). Large-scale resources such as Neuronpedia's circuit graphs operationalize these ideas in public tooling and highlight the need for methods that operate directly on CLT-native objects (Lindsey et al., 2025). Our method is designed explicitly for this setting: we score and prune CLT feature occurrences by aggregating their cross-layer writes, rather than reducing the problem back to neuron- or SAE-only views.

**Automated interpretation and evaluation of feature descriptions.** A complementary line of work aims to reduce human effort by generating interpretations of internal units automatically. Neuron- and feature-explainer systems

prompt an LLM with activation traces or exemplar contexts to propose a description of what a unit represents (Bills et al., 2023; Paulo et al., 2024; Lin, 2023). Output-centric and behavior-conditioned approaches push toward hypotheses tied more directly to model outputs, aiming to reduce purely correlational labels (Gur-Arieh et al., 2025). However, natural-language interpretations can be verbose, inconsistent, and difficult to falsify experimentally, motivating more structured or executable description formats (Huang et al., 2023). Recent work proposes semantic regex-style DSLs that produce compact, checkable descriptions and enable systematic evaluation through detection and fuzzing protocols (Boggust et al., 2025). In parallel, evaluation frameworks such as FADE formalize interpretation quality using metrics like clarity, purity, responsiveness, and faithfulness, including causal validation via patching (Puri et al., 2025). While these systems advance feature description quality, they typically assume a large pool of candidate units and do not address the upstream question of *which features should be explained*.

## 3. Method: The PIE Framework

We introduce **PIE** (Prune, Interpret, Evaluate), a CLT-native framework designed to resolve the efficiency bottleneck in mechanistic interpretability. While SAE features offer semantic sparsity, interpreting millions of features is computationally prohibitive. Our framework rests on the premise that *pruning must precede interpretation*: by identifying the sparse subset of Cross-Layer Transcoder (CLT) features causally relevant to a task, we can allocate interpretation budgets where they matter most.

The framework consists of three stages: (1) **Pruning** via Feature Attribution Patching (FAP) and its variants to select a minimal, high-fidelity circuit; (2) **Automatic Interpretation** of the retained features; and (3) **Evaluation** of both behavioral fidelity and explanation quality.

### 3.1. Problem Setup: CLT-Native Features

We analyze a subject model $M$ utilizing a Cross-Layer Transcoder (CLT) as a replacement model. Unlike standard transformers where edges connect modules (e.g., Attention Head $\rightarrow$ MLP), a CLT decomposes computation into sparse feature activations that write directly across layers.

Let a feature $f$ in the CLT dictionary have activation $a_f(x, t)$ at token position $t$ for input $x$. This feature contributes a vector $W_f^{(s)} \in \mathbb{R}^{d_{model}}$ to a downstream residual stream site $s$. We define the fundamental unit of analysis as the **feature-write edge**: the specific contribution of feature $f$ to site $s$ at position $t$. Pruning involves selecting a subset of features $\mathcal{F}^*$ (where $|\mathcal{F}^*| \ll |\mathcal{F}_{total}|$) such that the replacement model maintains low KL-divergence from the original

subject model $M$.

### 3.2. Pruning via Feature Attribution Patching (FAP)

To scale pruning to millions of features, we cannot afford iterative ablation (e.g., ACDC). Instead, we propose **Feature Attribution Patching (FAP)**, a first-order approximation method adapted for CLT feature space.

**Gradient-Weighted Write Attribution.** We estimate feature importance by combining activation differences with downstream gradients. Given a clean input $x_{\text{clean}}$ and a corrupted input $x_{\text{corr}}$, let $\Delta a_f(t) = a_f(x_{\text{clean}}, t) - a_f(x_{\text{corr}}, t)$ be the activation difference for feature $f$. We compute the gradient of a target metric $\mathcal{L}$ (e.g., logit difference or negative KL) with respect to the receiver site activation $h_s(t)$, denoted as $\nabla_{h_s(t)}\mathcal{L}$. The FAP score for feature $f$ is the dot product of its weighted write vector and the gradient:

$$S_{\text{FAP}}(f) = \sum_{t,s} \left( \Delta a_f(t) \cdot W_f^{(s)} \right)^{\top} \nabla_{h_s(t)}\mathcal{L} \qquad (1)$$

This score efficiently estimates how much feature $f$ contributes to restoring the clean behavior from the corrupted state. We retain the top-$K$ features with the highest magnitude scores $|S_{\text{FAP}}(f)|$. This acts as a rapid, low-cost filter to discard relevant noise.

### 3.3. Addressing Magnitude Failure Modes

While vanilla FAP is efficient, magnitude-based pruning suffers from two distinct failure modes in CLTs: *synergy ignorance* (missing features that only matter deeply in combination) and *sign cancellation* (discarding balancing inhibitory features). We introduce two specialized variants to address these.

**FAP-Synergy: Boundary Interaction Reranking.** Magnitude pruning often fails at the "pruning boundary" where the threshold separating kept and pruned features. Features just below the cutoff may be individually weak but highly synergistic with features already selected. FAP-Synergy addresses this via a **boundary reranking** procedure. We define a set of "Core" features $\mathcal{F}_{\text{core}}$ (highest scores) and "Boundary" candidates $\mathcal{F}_{\text{bound}}$ (scores near the cutoff). For a candidate $f_b \in \mathcal{F}_{\text{bound}}$ and a partner $f_c \in \mathcal{F}_{\text{core}}$, we estimate pairwise synergy via patching:

$$\text{Syn}(f_b, f_c) = \mathcal{M}(\{f_b, f_c\}) - \mathcal{M}(\{f_b\}) - \mathcal{M}(\{f_c\}) \quad (2)$$

where $\mathcal{M}$ represents the metric recovery (e.g., logit restoration) relative to the baseline. If a boundary feature exhibits strong positive synergy with core features, its score is boosted ($S'_f = S_f + \lambda \cdot \text{Syn}$), effectively rescuing high-information features that vanilla FAP would discard. This

method aims to lower the KL divergence of the pruned circuit without increasing the feature budget.

We additionally evaluate a sign-aware budget allocation variant (FAP-Signal) to mitigate sign cancellation; see Appendix D.

### 3.4. Automatic Interpretation & Evaluation

Once features are pruned, we generate natural language explanations using an LLM (e.g., GPT-5.2) prompted with max-activating exemplars (Bills et al., 2023). Crucially, because we interpret only the pruned set, we drastically reduce API costs compared to dense sweeps. We evaluate the quality of the discovery process using two categories of metrics.

**Behavioral Fidelity Metrics.** We measure how well the pruned feature set $\mathcal{F}^*$ preserves the original model mechanics:

- **KL Divergence:** The KL divergence between the subject model's output distribution and the replacement model restricted to $\mathcal{F}^*$, measured at the last token. Lower is better.

- **Prediction Change Rate:** The frequency with which the argmax token prediction changes after pruning.

**Interpretability Metrics.** To ensure that FAP selects meaningfully interpretable units (rather than just polysemantic error-correcting terms), we evaluate the generated explanations using the FADE framework metrics (Puri et al., 2025):

- **Clarity:** Can an independent auditor LLM generate synthetic samples that activate the feature based *only* on the description? (Measured via Gini coefficient of activations on synthetic vs. control data).

- **Purity:** When presented with real dataset examples, can the explanation distinguish high-activating samples from low-activating ones? (Measured via Average Precision).

- **Responsiveness:** Do natural samples that align with the description consistently trigger the feature? (Measured via Gini coefficient on rated natural samples).

We benchmark these metrics against a baseline of **1000 Random Features** to quantify the interpretability gain provided by our pruning strategies.

### 3.5. Practical Efficiency Considerations

FAP is practical at CLT scales: computing $S_{\text{FAP}}$ uses cached activations plus a small number of backward passes for

$g_s(t)$, and is typically far cheaper than iterative edge-search methods (e.g., ACDC (Conmy et al., 2023)). Synergy reranking is limited to a small boundary set to keep pairwise tests tractable, and the sign-aware variant (FAP-Signal; Appendix D) adds negligible overhead by splitting the budget by score sign.

# 4. Experiments

We evaluate the PIE framework on the Indirect Object Identification (IOI) task (Wang et al., 2023) to assess whether FAP-based pruning preserves behavioral fidelity while yielding features with high intrinsic interpretability.

## 4.1. Experimental Setup

**Models & Task.** We apply our framework to two open-weights models, **Gemma-2-2B** (Team et al., 2024) and **Llama-3.2-1B** (Grattafiori et al., 2024), utilizing pre-trained Cross-Layer Transcoders (CLTs) (Hanna et al., 2025). We target the IOI task (Wang et al., 2023), using a dataset of $N{=}2000$ prompts where the model must identify the indirect object. Detailed model specifications are provided in Appendix B.

**Pruning Baselines.** We compare our proposed FAP variants against a random baseline of 1000 features. All pruning methods select a fixed budget of $K{=}100$ features:

- **FAP (Base):** Selects top features by magnitude of gradient-weighted write attribution.

- **FAP-Synergy:** Reranks boundary features based on pairwise synergy estimation ($\lambda = 3$, top-25% boundary). The selection experiment of Hyperparameters for FAP-Synergy is detailed in Appendix A.

## 4.2. Evaluation Pipeline

To rigorously validate the PIE framework, we employ a three-stage protocol that separates the discovery distribution from the evaluation distribution:

**1. Pruning (Task-Specific).** We prune the CLT on the IOI dataset. We measure behavioral fidelity via **KL Divergence** and **Prediction Change Rate** (PCR) on the IOI validation set to quantify how well the sparse circuit reproduces the original model's task performance.

**2. Interpretation (Feature-Centric).** For the retained features, we generate natural language explanations using the Max-Act protocol (Bills et al., 2023). We utilize the pre-computed activation history from Circuit Tracer to retrieve max-activating exemplars efficiently.

**3. Evaluation (Generalization).** We evaluate the explanation quality (Clarity, Purity, Responsiveness) on a held-out

dataset of 2 million Wikipedia sentences (Sentence Transformers, 2024). Crucially, evaluating on Wikipedia rather than IOI or Circuit Tracer pre-computed activation history ensures that the explanations capture the features' general semantics, not just their task-specific utility. Implementation details for the scoring LLMs and sample sizes are detailed in Appendix B.

# 5. Results

We evaluate PIE on Llama-3.2-1B and Gemma-2-2B using the IOI task. We compare FAP variants ($K = 100$) against a baseline of randomly sampled features.

## 5.1. The Signal-to-Noise Interpretability Gap

A primary hypothesis of this work is that task-relevant features are inherently more interpretable than the general population. We quantify this by measuring the distribution of interpretation quality metrics (Clarity, Purity, Responsiveness).

**Reduction of Uninformative Units.** We formally define a feature as **uninformative** or "noisy" if it fails to meet a minimal quality threshold in either description generation or validation. Specifically, a feature is classified as uninformative if its Responsiveness $< 0.5$ AND Purity $< 0.5$.

As shown in Table 1, random sampling yields a high rate of such low-quality features (46.7% for Llama). FAP-based pruning significantly filters this noise, reducing the uninformative rate by approximately **9% absolute** (Llama) and **3.5% absolute** (Gemma). This confirms that PIE does not just select for magnitude; it systematically selects for semantic distinctness.

We report correlations and distributions among Clarity, Purity, and Responsiveness in Appendix C.

## 5.2. Semantic Efficiency and Signal Analysis

While standard metrics show parity, the true advantage of FAP-Synergy lies in its **efficiency**—the amount of interpretable signal retained per unit of behavioral degradation. Because Synergy actively repairs circuit boundaries to lower KL divergence, it minimizes the "cost" of pruning. To quantify this, we introduce three efficiency-oriented metrics:

- **Semantic Cost Efficiency (SCE):** A yield metric defined as $\frac{\text{Clarity+Purity}}{\text{KL·PCR}}$. This rewards methods that maximize semantics while strictly minimizing the joint compounded cost of model degradation.

- **Signal-to-Divergence Ratio (SDR):** A decibel-scale metric defined as $10 \cdot \log_{10}(\frac{\text{Responsiveness}}{\text{KL}})$. This separates the interpretable signal from the behavioral noise.

*Table 1.* Interpretability quality of FAP-pruned circuits ($K = 100$) vs. Random baseline. We report Mean $\pm$ Std for quality metrics. **Uninformative Rate** denotes the percentage of features where Responsiveness $< 0.5$ and Purity $< 0.5$. FAP-Synergy consistently maintains high interpretability while minimizing uninformative units.

| METHOD | CLARITY ($\uparrow$) | PURITY ($\uparrow$) | RESPONSIVENESS ($\uparrow$) | UNINFORMATIVE RATE ($\downarrow$) |
|---|---|---|---|---|
| *Llama-3.2-1B* | | | | |
| RANDOM (1K) | $0.585 \pm 0.320$ | $0.326 \pm 0.341$ | $0.455 \pm 0.364$ | 46.70% |
| FAP (BASE) | $0.627 \pm 0.032$ | $0.477 \pm 0.042$ | $0.465 \pm 0.041$ | 37.98% |
| FAP-SYNERGY | $\mathbf{0.627} \pm 0.032$ | $\mathbf{0.477} \pm 0.042$ | $\mathbf{0.465} \pm 0.041$ | **37.95%** |
| *Gemma-2-2B* | | | | |
| RANDOM (1K) | $0.606 \pm 0.317$ | $0.360 \pm 0.221$ | $0.475 \pm 0.255$ | 37.60% |
| FAP (BASE) | $0.635 \pm 0.036$ | $0.524 \pm 0.037$ | $0.534 \pm 0.041$ | 34.05% |
| FAP-SYNERGY | $0.635 \pm 0.036$ | $\mathbf{0.524} \pm 0.037$ | $\mathbf{0.535} \pm 0.041$ | **34.04%** |

- **Risk-Adjusted Responsiveness (RAR):** Defined as $\frac{\text{Responsiveness}}{\text{KL} \cdot \text{PCR}}$, isolating the responsiveness gain normalized by risk.

As shown in Table 2, FAP-Synergy dominates across all efficiency metrics. For Llama-3.2-1B, Synergy improves SCE by over **900 points** and RAR by over **500 points** compared to the Base method. This confirms that Synergy is not just selecting random features; it is selecting features that provide a higher return on interpretation investment by enforcing tighter behavioral bounds.

### 5.3. Behavioral Fidelity and Pruning Efficiency

We compare the behavioral fidelity of our pruned circuits. As shown in Table 3, FAP-Synergy achieves the best (lowest) KL Divergence on both models. For Llama, Synergy achieves a KL of 1.12 vs 1.13 for the Base method. While this margin appears small in linear terms, it represents a critical correction of circuit behavior, which we analyze further in Section 5.4.

**A Rigorous Random Baseline.** To ensure a fair comparison, our random baseline does not sample from the full CLT dictionary (which would be trivial, as most features are inactive). Instead, we sample $K$ features uniformly *only from the set of active features* for each specific prompt. For Llama-3.2-1B, the mean number of active feature occurrences ($N_{enc}$) per prompt is 4,188; for Gemma-2-2B, it is 5,190. Selecting from this active set constitutes a "strong" baseline: it guarantees that every randomly selected feature actually contributes non-zero information to the residual stream.

**The 40x Compression Gap.** We compare our methods at a strict budget of $K=100$ against the random baseline swept across various $K$ values (Figure 2).

- **At $K=100$:** The random baseline fails catastrophically. For Gemma-2-2B, random selection yields a KL

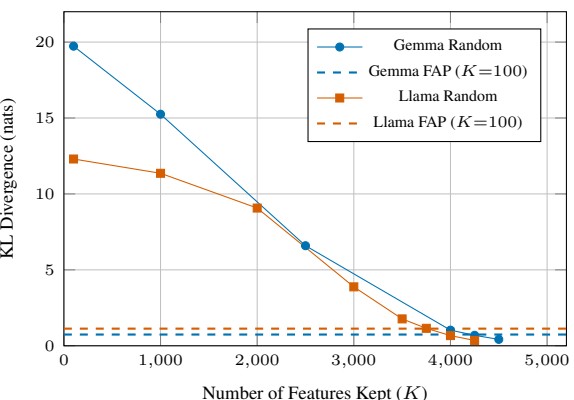

*Figure 2.* **Fidelity Efficiency Gap.** We plot the KL divergence of the Random baseline as the budget $K$ increases (solid lines). The dashed lines represent the KL achieved by FAP using only $K=100$ features. FAP achieves with 100 features what Random selection achieves with $\approx$4,000 features, demonstrating a compression factor of $\approx$40x on the active feature set.

of 19.73 and a PCR of 100%, indicating total model collapse. In contrast, FAP ($K=100$) achieves a low KL of 0.74 and a PCR of 58%.

- **Effective Compression:** To match the fidelity of FAP ($K=100$), the random baseline requires retaining approximately 4,250 features for Gemma (KL 0.69) and 3,750 features for Llama (KL 1.15).

This implies that FAP is approximately **40x more efficient** than random sampling even within the active feature set, which achieves comparable yield under a 40× smaller LLM/API evaluation budget, detailed in Appendix E.

### 5.4. Case Study: Rescuing Synergistic Components

To illustrate the mechanism of FAP-Synergy, we analyze a specific "boundary" feature in Llama-3.2-1B that would have been discarded by standard magnitude pruning but was successfully rescued by our interaction-aware reranking.

*Table 2.* **Semantic Efficiency Analysis.** By viewing KL and PCR as costs, we observe that FAP-Synergy yields significantly higher interpretability per unit of behavioral degradation. **SCE** measures total semantic yield, **SDR** measures signal-to-noise in dB (closer to 0 is better), and **RAR** measures responsiveness relative to risk. Best results are bolded.

| MODEL | METRIC | FAP (BASE) | FAP-SYNERGY | RANDOM |
|---|---|---|---|---|
| LLAMA-3.2-1B | SCE (EFFICIENCY) | 947,041 | **948,036** | 0.08 |
| | SDR (SIGNAL/NOISE) | -4.037 DB | **-4.032 DB** | -14.32 DB |
| | RAR (RESPONSIVENESS) | 394,682 | **395,188** | 0.04 |
| GEMMA-2-2B | SCE (EFFICIENCY) | 1,620,789 | **1,621,365** | 0.05 |
| | SDR (SIGNAL/NOISE) | -1.223 DB | **-1.221 DB** | -16.19 DB |
| | RAR (RESPONSIVENESS) | 754,563 | **754,931** | 0.02 |

*Table 3.* **Comprehensive Fidelity Analysis.** We report the KL Divergence and Prediction Change Rate (PCR) for the Random baseline across a wide sweep of feature budgets ($K$), compared to our FAP methods at a strict budget of $K=100$. While Random selection requires $K \approx 4000$ to achieve acceptable fidelity, FAP variants achieve comparable or better performance with only 100 features.

| METHOD | BUDGET ($K$) | KL ($\downarrow$) | PCR ($\downarrow$) |
|---|---|---|---|
| *Llama-3.2-1B* | | | |
| RANDOM | 100 | 12.30 | 96.7% |
| RANDOM | 1000 | 11.37 | 97.6% |
| RANDOM | 2000 | 9.07 | 97.2% |
| RANDOM | 3000 | 3.88 | 68.7% |
| RANDOM | 3500 | 1.78 | 38.6% |
| RANDOM | 3750 | 1.15 | 30.8% |
| RANDOM | 4000 | 0.67 | 24.2% |
| RANDOM | 4250 | 0.35 | 15.8% |
| **FAP (BASE)** | **100** | **1.13** | **44.1%** |
| **FAP-SYNERGY** | **100** | **1.12** | **44.1%** |
| *Gemma-2-2B* | | | |
| RANDOM | 100 | 19.73 | 100.0% |
| RANDOM | 1000 | 15.25 | 99.5% |
| RANDOM | 2500 | 6.60 | 90.6% |
| RANDOM | 4000 | 1.03 | 44.3% |
| RANDOM | 4250 | 0.69 | 35.0% |
| RANDOM | 4500 | 0.42 | 24.3% |
| **FAP (BASE)** | **100** | **0.74** | **58.3%** |
| **FAP-SYNERGY** | **100** | **0.73** | **58.3%** |

**The Hidden Architect: Orthography meets Semantics.** We examine **Feature L0.2703**, interpreted as a "J-initial token detector" (activates on "*J*", "*Java*", "*Jacob*"). Under base FAP, this feature fell slightly below the Top-$K$ threshold ($S_{\text{FAP}}$ magnitude ranking) and was slated for removal. However, FAP-Synergy identified strong pairwise interactions with retained "Core" features, boosting its score to safe retention.

**Positive Synergy: Constructive Amplification.** Our analysis shows that L0.2703 provides crucial upstream support for higher-level semantic features. In particular, we identify **L5.8201** (*"Given Name"* detector) as the **8th strongest**

**positive-synergy partner** of L0.2703 (Synergy $\approx +0.156$): the model appears to use the low-level orthographic cue (*"starts with J"*) from Layer 0 to amplify the confidence of the mid-level semantic signal (*"is a name"*) at Layer 5. As a result, when both features are present, the model's handling of names like ``Jacob'' exceeds what either feature achieves alone, whereas pruning the L0 trigger makes the L5 detector less reliable on this subset of names.

**Negative Synergy: Managing Redundancy.** Conversely, the framework also identified **Feature L0.5905** (specific "Jacob" detector) as the **1st highest negative synergy partner** ($Synergy \approx -0.375$). This strong negative interaction indicates redundancy: the specific ``Jacob'' feature and the general ``J-initial'' feature likely encode overlapping evidence for the token ``Jacob''. From an interpretability standpoint, this suggests either that L0.5905 is an overfitted backup of L0.2703 introduced during CLT reconstruction, or that activating L0.5905 creates a shortcut that suppresses L0.2703's contribution in the Jacob context. By rescuing L0.2703, PIE preserves the circuit's *compositional* structure (Orthography $\rightarrow$ Semantics), rather than yielding a disjoint set of isolated semantic detectors.

## 6. Discussion

**PIE makes CLT interpretability *budgetable*.** A central obstacle in end-to-end circuit analysis is not discovering candidate units, but deciding *which* units are worth spending scarce interpretation budget on. PIE reframes this as a *budgeted pipeline*: (i) prune a large candidate set down to a small subset that preserves behavior, (ii) spend expensive auto-interpretation and evaluation only on that subset, and (iii) report both behavioral fidelity and interpretability outcomes under a fixed budget. This is practically important because explanation and evaluation are the dominant cost in modern interpretability workflows (LLM-based description, counterfactual prompting, and FADE-style scoring) (Puri et al., 2025; Boggust et al., 2025). By turning "interpret everything" into "interpret the *right K*," PIE enables systematic comparisons across methods and models under a consistent interpretation budget.

**Why the $\sim$40$\times$ compression gap matters.** Even when random baselines sample from the active feature set, they require thousands of features to match the behavioral preservation achieved by PIE with $K=100$. This gap indicates that the behaviorally relevant computation is concentrated in a much smaller *causal core*, while many active features are redundant, weakly coupled, or act as backups that do not materially affect the target behavior. From a systems perspective, this compression is the primary driver of feasibility: reducing the interpretation target from $\mathcal{O}(10^3–10^4)$ to $\mathcal{O}(10^2)$ features directly translates to large savings in LLM calls, tokens, and human inspection time. From a scientific perspective, it sharpens mechanistic hypotheses: if the circuit can be summarized by a small set of features, we can more plausibly map them to functional roles and test those roles causally.

**The relatively "small" KL win from synergy reranking.** Synergy-aware reranking yields modest but consistent improvements in behavioral fidelity relative to magnitude-only pruning. It should be noted that this regime is intentionally *boundary-local*: reranking only perturbs the final selection among near-threshold candidates, so large KL swings are neither expected nor desirable. The value of synergy reranking is therefore not solely in absolute KL reduction, but in improving the *interpretability–fidelity tradeoff*: among similarly faithful pruned sets, synergy tends to favor feature combinations that better preserve coordinated computation (e.g., complementary roles rather than redundant backups). This also motivates synergy diagnostics (e.g., sign- and interaction-aware statistics) as an analysis tool: they offer a handle on when two features jointly matter beyond their marginal contributions, which magnitude-based criteria systematically ignore.

**Mechanistic lessons from the case study.** The L0.2703 case study illustrates how PIE supports *mechanistic* claims rather than only aggregate metrics. First, it highlights that circuits can be compositional across layers: features may play distinct roles (e.g., orthographic or structural cues upstream, semantic selection downstream) that must be preserved jointly to maintain behavior. Second, it exposes failure modes of magnitude-only selection: highly active features can function as "backup" or "shortcut" pathways that partially compensate for other features but obscure the true causal decomposition. By surfacing both positive and negative interaction partners, the synergy view provides qualitative evidence that the retained set is not merely a collection of strong individual detectors, but a coordinated subset that better matches the circuit's functional structure. These case studies are thus best read as *mechanistic validation* that complements the global compression and fidelity results.

**Synergy may be under-measured by "feature-isolated" evaluation.** The case study suggests that synergy-aware reranking's effect is *structural*: it preferentially retains *complementary* features whose joint presence preserves coordinated computation, rather than substitutable "backup" features that match behavior only marginally. In other words, synergy helps preserve *interactions*, in which the way features compose into a circuit, even when their marginal, feature-by-feature contributions look similar at the pruning boundary. This distinction may be weakly expressed in our present evaluation setting because our interpretability pipeline largely treats features as *isolated* units (explained and scored independently), which is not fully "circuit-friendly" and can under-reward interaction-preserving selections. The mechanistic lessons therefore motivate an important direction for future work: developing a *CLT-native* evaluation pipeline that explicitly evaluates *sets* of features and their composition (e.g., interaction-aware interventions, group-level counterfactuals, and structure-sensitive explanation scoring). Such an evaluation would better capture the value of synergy-preserving selection and enable a clearer understanding of when and why synergy improves circuit faithfulness beyond what marginal metrics reveal.

# 7. Conclusion

We introduced PIE, the first end-to-end CLT-native interpretability framework that makes mechanistic analysis *budgetable* by explicitly connecting feature pruning, automatic interpretation, and FADE-style evaluation under a fixed interpretation budget. At the core of PIE is Feature Attribution Patching (FAP), a patch-grounded, gradient-weighted write attribution method that rapidly filters CLT feature space, together with FAP-Synergy, which improves boundary selection by reranking near-threshold features using pairwise interaction signals.

Empirically, on IOI with CLTs for Llama-3.2-1B and Gemma-2-2B, PIE isolates a sparse "causal core": retaining only $K=100$ features achieves behavioral fidelity that a strong random baseline requires *thousands* of active features to match, yielding an approximate $\sim$**40**$\times$ compression even within the active feature set and correspondingly large reductions in downstream interpretation/evaluation calls. Synergy-aware reranking produces modest but consistent fidelity gains at fixed budget while preserving the compositional structure of circuits by rescuing features that matter primarily through interactions. Looking forward, these results motivate more **circuit-aware** evaluation protocols that score *sets* of interacting features (rather than isolated units), as well as broader scaling studies across tasks, budgets, and CLT training regimes.

## Impact Statement

Our work aims to make mechanistic interpretability scalable and economically viable. By enabling the pruning of Cross-Layer Transcoder (CLT) features, the PIE framework significantly reduces the computational and financial costs associated with automated model interpretation. This advances the goal of AI transparency, making it feasible to audit large models for safety-critical behaviors without analyzing millions of redundant parameters. Furthermore, by lowering the resource requirements for circuit discovery, this work promotes inclusivity in the research community, allowing entities with smaller compute budgets to participate in safety research. However, we acknowledge the risk of over-reliance on pruned circuits; aggressive pruning might obscure subtle, distributed computations necessary for full behavioral robustness. Researchers should treat pruned interpretations as high-signal approximations rather than exhaustive descriptions of model psychology.

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

## A. Hyperparameter Selection for FAP-synergy

**Baseline.** We use the ordinary FAP setting (synergy weight $\lambda = 0$) as the baseline on IOI with $N{=}500$, $K{=}100$. The baseline metrics are: mean last-token KL = 1.1399, std = 0.5393, prediction-change rate = 0.446. All results below are reported as *deltas relative to this baseline*.

**Selection rule.** We select hyperparameters by *minimizing mean last-token KL*. Even sub-percent improvements are meaningful in interpretability pruning studies, where improvements are typically incremental yet consistent across large prompt sets.

### A.1. Sensitivity Analysis

We conducted a comprehensive sweep over the synergy weight $\lambda \in \{1, \ldots, 5\}$ and boundary percent $bp \in \{20, \ldots, 45\}$ to identify the optimal configuration for FAP-synergy. The results are summarized in Table 4 and visualized in Figure 3.

As shown in the table, the global optimum (lowest $\Delta$ mean KL) is achieved at $\lambda = 3$ with a boundary percent of $bp = 25$. The data reveals a consistent trend across all tested $\lambda$ values: increasing the boundary percent beyond $bp = 25$ degrades performance (e.g., at $bp = 40$, the improvement drops significantly to $\approx -0.758$ milli-KL compared to $> -1.0$ at $bp = 25$). This suggests that widening the reranking window too far introduces noise or dilutes the high-synergy pairs with less relevant features. Furthermore, increasing $\lambda$ to 4 or 5 yields nearly identical results to $\lambda = 3$ at the optimal $bp$, but does not surpass it. Consequently, we select $\lambda = 3, bp = 25$ as the most robust configuration.

### A.2. Delta table (vs. baseline)

### A.3. Scatter plot (delta vs. baseline)

## B. Implementation Details

### B.1. Models and Resources

We utilize the following public checkpoints:

- **Gemma-2-2B** **CLT:** mntss/clt-gemma-2-2b-426k (Dictionary size: 426k).

- **Llama-3.2-1B** **CLT:** mntss/clt-llama-3.2-1b-524k (Dictionary size: 524k).

All pruning operations are performed on a single NVIDIA A100 GPU. The IOI dataset is generated using the template structure from Wang et al. (2023), consisting of 2000 unique prompts with varying names and objects.

*Table 4.* Sweep results reported as deltas vs. the $\lambda = 0$ baseline (negative is better). $\Delta$KL and $\Delta$std are shown in *milli-KL* (i.e., $\times 10^3$) for readability. The selected setting is $\lambda = 3$, bp= 25.

| $\lambda$ | bp | $\Delta$mean KL ($\times 10^3$) $\downarrow$ | $\Delta$std ($\times 10^3$) $\downarrow$ |
|---|---|---|---|
| 1 | 20 | -0.270 | -0.353 |
| 1 | 25 | -0.785 | -0.770 |
| 1 | 30 | -0.504 | -0.349 |
| 1 | 35 | -0.668 | -0.509 |
| 1 | 40 | -0.758 | -0.452 |
| 1 | 45 | -0.203 | -0.165 |
| 2 | 20 | -0.074 | -0.293 |
| 2 | 25 | -0.887 | -0.654 |
| 2 | 30 | -0.488 | -0.334 |
| 2 | 35 | -0.645 | -0.506 |
| 2 | 40 | -0.758 | -0.452 |
| 2 | 45 | -0.203 | -0.165 |
| 3 | 20 | -0.035 | -0.303 |
| **3** | **25** | **-1.066** | **-1.032** |
| 3 | 30 | -0.504 | -0.339 |
| 3 | 35 | -0.645 | -0.506 |
| 3 | 40 | -0.758 | -0.452 |
| 3 | 45 | -0.203 | -0.165 |
| 4 | 20 | -0.035 | -0.303 |
| 4 | 25 | -1.043 | -1.015 |
| 4 | 30 | -0.504 | -0.339 |
| 4 | 35 | -0.645 | -0.506 |
| 4 | 40 | -0.758 | -0.451 |
| 4 | 45 | -0.203 | -0.165 |
| 5 | 20 | -0.027 | -0.303 |
| 5 | 25 | -1.043 | -1.015 |
| 5 | 30 | -0.488 | -0.323 |
| 5 | 35 | -0.645 | -0.506 |
| 5 | 40 | -0.758 | -0.451 |
| 5 | 45 | -0.203 | -0.165 |

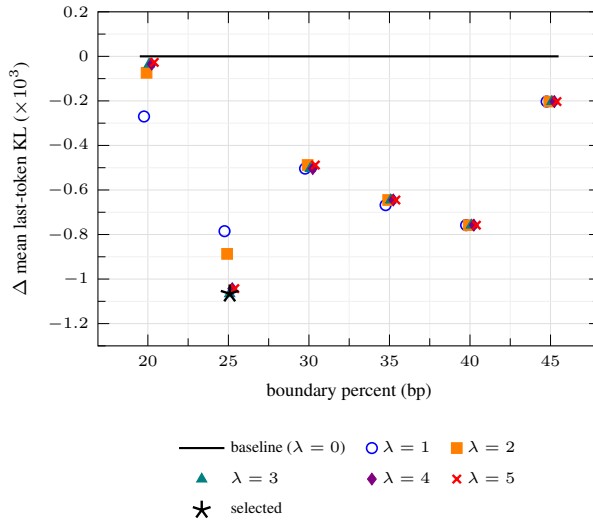

*Figure 3.* Sweep scatter plot reported as $\Delta$ mean KL vs. baseline ($\lambda = 0$).

### B.2. Interpretation and Evaluation Protocol

**Step 2: Interpretation Generation.** We use `gpt-5.2` as the explainer model.

- **Exemplars:** We provide 40 max-activating examples per feature, retrieved from the Circuit Tracer activation cache.

- **Highlighting:** Tokens are highlighted if their activation value exceeds 65% of the max activation in the sequence.

**Step 3: FADE Evaluation.** We use `gpt-5o-mini` as the auditor model to compute the metrics defined in Section 3.4.

- **Evaluation Dataset:** We draw samples from the (Sentence Transformers, 2024) dataset.

- **Clarity:** The auditor generates 15 synthetic positive and negative examples based solely on the description. We measure the Gini coefficient of the feature's activations on these synthetic batches.

- **Purity and Responsiveness:** We evaluate the feature on a retrieved set of $N_{eval} = 250$ real examples from

Wikipedia (distinct from the exemplar set). Purity is calculated as the Average Precision (AP) of the description acting as a binary classifier for feature activity. Responsiveness is measured as the activation difference between samples rated as "relevant" vs "irrelevant" by the auditor.

## C. Metric Correlations and Distributions

In this section, we analyze the distribution of feature quality scores and the relationship between behavioral fidelity and interpretability.

**Signal-to-Noise Gap.** As shown in Figure 6, there is a distinct "signal-to-noise" gap between random sampling and our method. Random features cluster near zero across all metrics, whereas FAP variants consistently select features in the high-interpretability regime.

**Feature Correlations.** We investigate the relationship between Clarity, Purity, and Responsiveness within the FAP-selected set. Our analysis reveals a robust positive correlation between Responsiveness and Purity, with Pearson $r \approx 0.71$ for Gemma-2-2B and $r \approx 0.73$ for Llama-3.2-1B (Figure 4). This strong correlation suggests that PIE selects features with jointly high Purity and Responsiveness, i.e., concept-specific activations that remain stable across natural data; FADE notes that when descriptions/features are well aligned, interpretability metrics tend to rise together (Puri et al., 2025).

**Prompt Fidelity vs. Interpretability.** We further ask whether "better" circuits (those with lower KL divergence)

yield more interpretable features. We compute the Pearson correlation between the prompt-level KL divergence and the mean interpretability scores of the retained features (Figure 5). For **Clarity**, we observe a significant negative correlation (Llama $r \approx -0.36$, Gemma $r \approx -0.17$; $p < 0.001$). Since lower KL indicates better fidelity, this result implies that **prompts where the pruned circuit functions well tend to contain features that are easier to explain**. Interestingly, Purity and Responsiveness show a weaker or slightly inverse relationship with KL (Llama $r \approx 0.15$ for Purity), suggesting that while functional fidelity strongly predicts the *clarity* of the mechanism, the *monosemanticity* (purity) of features may depend more on the specific semantic content of the prompt (e.g., presence of specific named entities) rather than the circuit's overall error rate.

## D. FAP-Signal: Sign-Aware Budget Allocation (Ablation)

**Motivation: sign cancellation in magnitude pruning.** Magnitude-based attribution can under-select features with negative (inhibitory) contributions when positive features dominate the score distribution. In mechanistic circuits, such "brake" components may be necessary to preserve calibrated behavior and to prevent systematic over-correction under pruning.

**Definition.** Let $S_{\text{FAP}}(f)$ be the base Feature Attribution Patching score (Eq. (1)). FAP-Signal enforces a minimum allocation to both positive- and negative-scoring features at a fixed budget $K$. For a user-chosen ratio $\gamma \in (0, 0.5)$:

1. Select the top $\gamma K$ features from $\{f : S_{\text{FAP}}(f) > 0\}$ by score magnitude.

2. Select the top $\gamma K$ features from $\{f : S_{\text{FAP}}(f) < 0\}$ by score magnitude.

3. Fill the remaining $(1 - 2\gamma)K$ slots by the largest $|S_{\text{FAP}}(f)|$ among all remaining features.

This procedure preserves a signed "floor" of inhibitory features while retaining the simplicity and speed of the base top-$K$ selection.

**Experimental setting.** We evaluate FAP-Signal under the same protocol as the main paper: IOI pruning with $K{=}100$, followed by automated interpretation and FADE-style evaluation on a held-out Wikipedia distribution (Appendix B). We use $\gamma = 0.25$ throughout.

**Empirical outcome on IOI.** In our IOI experiments, FAP-Signal does not yield a consistent improvement over base FAP in either behavioral fidelity (KL, PCR) or interpretability metrics (Clarity, Purity, Responsiveness). Table 5 shows

*Table 5.* **Fidelity ablation for FAP-Signal.** Same setup as Table 3 in the main text, but including the sign-aware variant.

| MODEL | METHOD | KL ($\downarrow$) | PCR ($\downarrow$) |
|---|---|---|---|
| 3*LLAMA-3.2-1B | FAP (BASE) | 1.13 | 44.1% |
| | FAP-SIGNAL | 1.13 | 44.1% |
| | FAP-SYNERGY | 1.12 | 44.1% |
| 3*GEMMA-2-2B | FAP (BASE) | 0.74 | 58.3% |
| | FAP-SIGNAL | 0.74 | 58.3% |
| | FAP-SYNERGY | 0.73 | 58.3% |

that FAP-Signal matches base FAP to within noise on KL/PCR. Table 6 reports the three semantic-efficiency metrics introduced in Sec. 5.2: Semantic Cost Efficiency (SCE), Signal-to-Divergence Ratio (SDR), and Risk-Adjusted Responsiveness (RAR). Across both models, FAP-Signal does not improve over base FAP and remains slightly below FAP-Synergy.

**Interpretation.** These results suggest that, for IOI on the evaluated CLTs, sign cancellation is not the dominant failure mode of base FAP at $K{=}100$. In contrast, boundary interactions (synergy) appear to be the more salient source of pruning error, motivating the main focus on FAP-Synergy.

## E. Economic Analysis

Our PIE pipeline incurs API cost only in the *Interpret* and *Evaluate* stages. Under our experimental configuration, the **explanation** phase (GPT-5.2) consumes approximately 4,000 input tokens (system prompt + 40 max-activating exemplars) and produces $\approx 200$ output tokens *per feature*. The subsequent **evaluation** phase (GPT-5 mini, used for both synthetic clarity generation and purity rating) is more data-intensive, averaging 22,650 input tokens and 4,000 output tokens per feature. Under standard pricing(OpenAI, 2026), this yields a total estimated cost of

$$c_{\text{feat}} \approx \$0.0235 \quad \text{per interpreted feature.}$$

**Prompt-level budgeting (active set vs. $K{=}100$).** A rigorous alternative to interpreting *all* CLT features is to interpret only those that are *active* for the prompt. In our setting, the mean number of active feature occurrences per prompt is 4,188 for Llama-3.2-1B and 5,190 for Gemma-2-2B.

Interpreting the full active set for a *single* prompt would therefore cost

$$\text{Cost}_{\text{Llama, active/prompt}} \approx 4{,}188 \cdot c_{\text{feat}} \approx \$98.42, \quad (3)$$

$$\text{Cost}_{\text{Gemma, active/prompt}} \approx 5{,}190 \cdot c_{\text{feat}} \approx \$121.97. \quad (4)$$

In contrast, PIE interprets only a fixed budget of $K{=}100$ features, costing

$$\text{Cost}_{K=100} \approx 100 \cdot c_{\text{feat}} \approx \$2.35. \quad (5)$$

*Table 6.* **Efficiency metrics including FAP-Signal (restricted to metrics defined in the main text).** We report Semantic Cost Efficiency (SCE), Signal-to-Divergence Ratio (SDR; dB, closer to 0 is better), and Risk-Adjusted Responsiveness (RAR) as defined in Sec. 5.2. Across both models, FAP-Signal does not improve over base FAP, while FAP-Synergy provides the best overall efficiency.

| MODEL | METRIC | FAP (BASE) | FAP-SYNERGY | FAP-SIGNAL | RANDOM |
|---|---|---|---|---|---|
| 3*LLAMA-3.2-1B | SCE (EFFICIENCY) | 947,041.9409 | **948,036.9102** | 945,756.8388 | 0.0766 |
| | SDR (SIGNAL/NOISE) | -4.0375 DB | **-4.0320 DB** | -4.0431 DB | -14.3187 DB |
| | RAR (RESPONSIVENESS) | 394,682.5462 | **395,188.0711** | 394,175.8208 | 0.0383 |
| 3*GEMMA-2-2B | SCE (EFFICIENCY) | 1,620,789.2094 | **1,621,365.9427** | 1,620,789.2094 | 0.0490 |
| | SDR (SIGNAL/NOISE) | -1.2230 DB | **-1.2209 DB** | -1.2230 DB | -16.1878 DB |
| | RAR (RESPONSIVENESS) | 754,563.0057 | **754,931.0929** | 754,563.0057 | 0.0241 |

Thus, even when comparing against the *strong* baseline that restricts attention to active features, PIE reduces interpretation/evaluation spend by a factor of $\frac{4,188}{100} \approx 41.9\times$ (Llama) and $\frac{5,190}{100} \approx 51.9\times$ (Gemma), aligning with the observed $\approx 40\times$ fidelity efficiency gap on the active set.

**Dataset-level budgeting (global reuse across 2,000 prompts).** The prompt-level view is conservative because it does not exploit reuse: across a dataset, the same features appear repeatedly, so we can cache interpretations and only pay once per *unique* feature. Concretely, when aggregating across $N$=2000 prompts, the number of *unique* features that ever appear as `kept` after pruning is approximately 4,400 for Llama and 4,000 for Gemma. Under this global accounting, the total interpretation+evaluation cost becomes

$$\text{Cost}_{\text{Llama, global kept}} \approx 4,400 \cdot c_{\text{feat}} \approx \$103.40, \quad (6)$$

$$\text{Cost}_{\text{Gemma, global kept}} \approx 4,000 \cdot c_{\text{feat}} \approx \$94.00. \quad (7)$$

By comparison, a naive global sweep that attempts to interpret the *entire* CLT dictionary would require evaluating

$$\begin{aligned} |\mathcal{F}_{\text{Llama}}| &= 16 \times 32,768 = 524,288, \\ |\mathcal{F}_{\text{Gemma}}| &= 26 \times 16,384 = 425,984. \end{aligned} \quad (8)$$

which would cost

$$\text{Cost}_{\text{Llama, full dict}} \approx 524,288 \cdot c_{\text{feat}} \approx \$12,320.77, \quad (9)$$

$$\text{Cost}_{\text{Gemma, full dict}} \approx 425,984 \cdot c_{\text{feat}} \approx \$10,010.62. \quad (10)$$

Therefore, from a global perspective PIE reduces the evaluation burden by $\frac{524,288}{4,400} \approx 119\times$ on Llama and $\frac{425,984}{4,000} \approx 106\times$ on Gemma, translating to savings on the order of $\sim\$12.2\text{k}$ and $\sim\$9.9\text{k}$ for a single end-to-end run.

## F. Limitations

Our results should be interpreted in light of several practical limitations.

**Budgeted evaluation at a single feature budget.** To keep the end-to-end pipeline tractable, all of our pruning methods operate at a fixed circuit budget of $K$=100 retained features. As a result, we do not report a systematic scaling study over larger budgets (e.g., $K \in \{200, 500, 1000\}$) for our main pipeline, even though the random baseline is swept over $K$ in Figure 2 for context. While our conclusions are strongest in the strict-budget regime that motivated PIE, the relative ordering and the absolute interpretability metrics may change as $K$ increases.

**No CLT training; reliance on public replacement checkpoints.** We do not train Cross-Layer Transcoders (CLTs) ourselves; instead, we use public CLT checkpoints released with Circuit Tracer tooling (Gemma-2-2B CLT and Llama-3.2-1B CLT). Consequently, our claims are conditional on the quality and representational coverage of these replacement models. In particular, pruning behavior and feature semantics may differ for CLTs trained with different data, objectives, sparsity regimes, or architectures.

**Scope of causal claims.** PIE is designed to isolate a minimal circuit that reproduces behavior under the replacement-model intervention, but this does not automatically imply that every retained feature corresponds to a unique mechanistic "part" in a human-interpretable decomposition. Feature redundancy, polysemanticity, and interaction effects can persist even after pruning, and the replacement-model abstraction may miss mechanisms not captured by the CLT basis.

| Scenario | #Features | Cost | vs. Budgeted | Reduction |
|---|---|---|---|---|
| *Prompt-level (single prompt)* | | | | |
| Llama active set | 4,188 | $98.42 | $2.35 ($K{=}100$) | 41.9× |
| Gemma active set | 5,190 | $121.97 | $2.35 ($K{=}100$) | 51.9× |
| *Global (2,000 prompts; unique features)* | | | | |
| Llama kept (unique) | 4,400 | $103.40 | full dict: $12,320.77 | 119× |
| Gemma kept (unique) | 4,000 | $94.00 | full dict: $10,010.62 | 106× |

*Table 7.* **Economic impact of budgeting.** Using $c_{\text{feat}} \approx \$0.0235$ per feature, a fixed interpretation budget (and global reuse across prompts) yields large cost reductions relative to interpreting the full active set per prompt or sweeping the entire CLT dictionary.

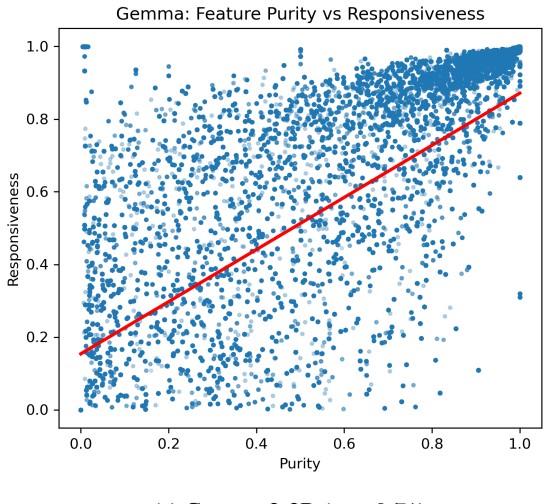

(a) Gemma-2-2B ($r \approx 0.71$).

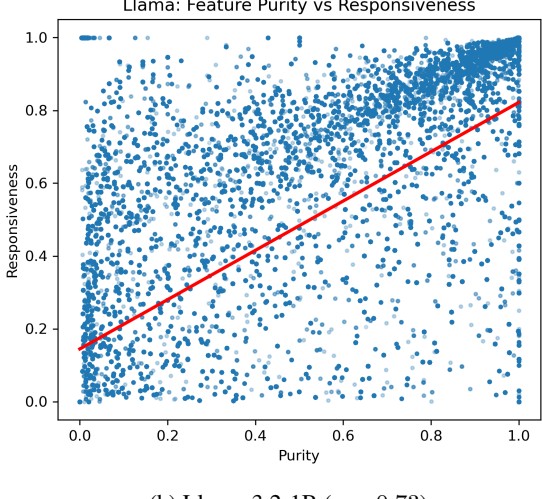

(b) Llama-3.2-1B ($r \approx 0.73$).

*Figure 4.* **Feature Purity vs. Responsiveness.** Scatter plots for FAP-selected features. We observe a strong positive correlation ($r > 0.7$) across both models, confirming that PIE selects features that are both precise (Pure) and sensitive (Responsive).

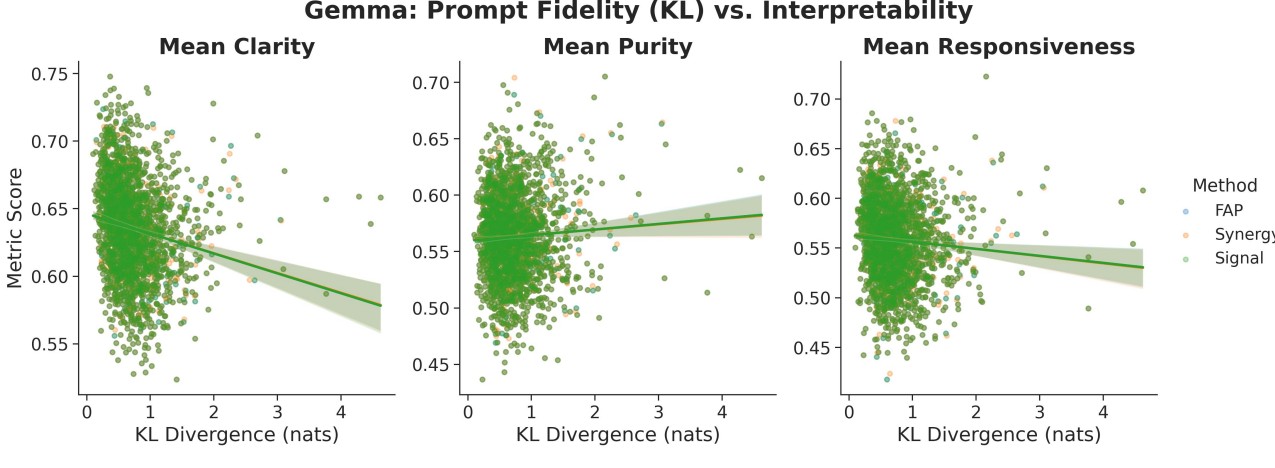

(a) Gemma-2-2B: Prompt Fidelity vs. Quality.

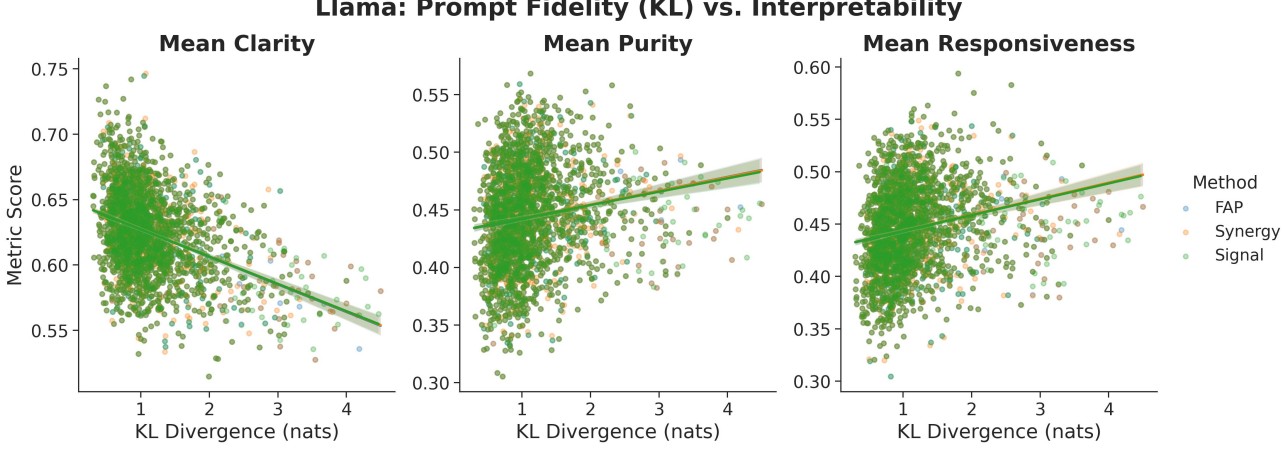

(b) Llama-3.2-1B: Prompt Fidelity vs. Quality.

*Figure 5.* **Does Fidelity Predict Interpretability?** We plot the KL Divergence of each prompt (x-axis, lower is better) against the mean interpretability metrics of the pruned features (y-axis). For *Mean Clarity* (left panels), we observe a consistent negative slope, indicating that circuits with higher behavioral fidelity (lower KL) are composed of features that are easier to explain.

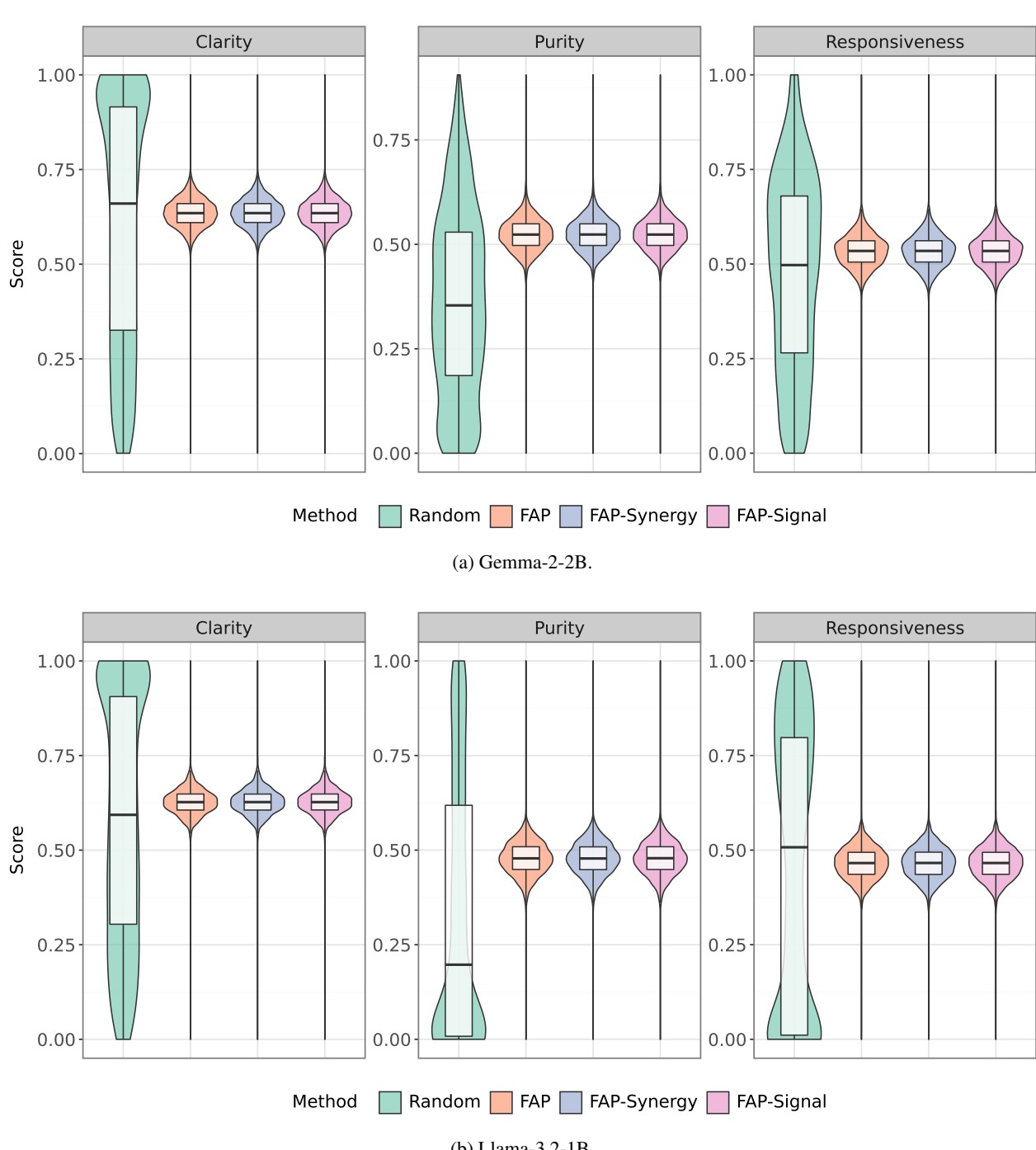

(a) Gemma-2-2B.

(b) Llama-3.2-1B.

*Figure 6.* **The Signal-to-Noise Gap.** Violin plots comparing Clarity, Purity, and Responsiveness for Random vs. FAP variants. Random sampling includes many low-quality features (scores near 0), while FAP selects features in the high-interpretability regime with low variance.

