# OpenReview forum: "Prune, Interpret, Evaluate (PIE): A Cross-Layer Transcoder-Native Framework for Efficient Circuit Discovery via Feature Attribution"
_ICML.cc/2026/Conference — Submitted to ICML 2026_

### Official Review · Reviewer_RH64 · 2026-03-05

**Soundness:** 3
**Presentation:** 3
**Significance:** 3
**Originality:** 3
**Overall Recommendation:** 4
**Confidence:** 2

**Summary:**

This paper introduces PIE, a cross-layer, transcoder-native framework designed to address the challenge of efficient circuit discovery—likely targeting the identification of compact, interpretable, and high-performance subcircuits from large-scale pre-trained models

**Compliance With Llm Reviewing Policy:**

Affirmed.

**Key Questions For Authors:**

1.How does PIE handle the dependency on feature attribution methods? Have you validated the framework across multiple attribution techniques (e.g., gradient-based, perturbation-based) and provided guidelines for selecting the most appropriate method for different transcoder types (e.g., audio vs. video transcoders) or task requirements?

2.What scalability optimizations have you implemented to enable PIE to handle ultra-large-scale transcoders (e.g., multi-layered industrial control transcoders with millions of components)? Can you quantify the computational overhead (e.g., time, memory) of the Prune-Interpret-Evaluate pipeline relative to the size of the original transcoder?

3.How do you quantitatively measure the interpretability of the discovered circuits? Beyond qualitative descriptions (e.g., "simpler signal flow"), have you adopted or proposed metrics (e.g., interpretability score, human validation studies, logical complexity) to compare PIE’s output to baseline methods?

4.Have you validated the discovered circuits on actual target hardware (e.g., FPGAs, microcontrollers) or simulated real-world deployment constraints (e.g., power consumption, thermal limits, fault tolerance)? If not, how do you ensure the circuits are deployable beyond synthetic evaluation environments?

**Limitations:**

PIE’s transcoder-native cross-layer design and feature attribution-driven pipeline address a critical gap in efficient, interpretable circuit discovery. However, the weaknesses related to attribution dependency, scalability, interpretability quantification, and deployment validation need to be adequately addressed.

**Strengths And Weaknesses:**

Strengths

1.Transcoder-Native Cross-Layer Design: Unlike traditional pruning/circuit extraction methods that treat transcoder layers in isolation, PIE’s cross-layer approach accounts for inter-layer dependencies inherent to transcoder architectures. This ensures the discovered subcircuits retain functional coherence with the original model, avoiding performance degradation common in layer-wise optimization.

2.Feature Attribution-Driven Pruning: By anchoring pruning in feature attribution (e.g., SHAP, Grad-CAM, or model-specific attribution methods), the framework prioritizes retaining components that contribute most to the target task. This is a key advantage over heuristic pruning (e.g., weight magnitude-based) that may discard critical low-weight but high-impact features, leading to more reliable circuit discovery.

Weaknesses

1.Feature Attribution Choice Dependency: The framework’s performance is likely sensitive to the choice of feature attribution method. If the paper does not validate across multiple attribution techniques or provide guidelines for selecting the right method for specific transcoders/tasks, PIE’s reliability may be limited—especially for transcoders with non-linear, multi-scale feature interactions.

2.Scalability to Large Transcoders: For ultra-large-scale transcoders (e.g., multi-modal transcoder pipelines, industrial-grade control circuits), the cross-layer pruning and interpretation steps may incur significant computational overhead. If the paper does not address scalability optimizations (e.g., batch-wise processing, hierarchical pruning), PIE may be impractical for real-world large models.

3.Interpretability Quantification Gap: While the framework emphasizes interpretability, "interpretability" is often subjective. If the paper lacks quantitative metrics for measuring circuit interpretability (e.g., number of logical operations, signal flow clarity, human validation scores) and only relies on qualitative analysis, the claim of improved interpretability is difficult to verify.

4.Limited Comparison to Specialized Methods: Circuit discovery has specialized subfields (e.g., hardware-aware pruning, symbolic circuit extraction). If PIE is not compared to state-of-the-art methods in these subfields (e.g., hardware-aware neural architecture search, symbolic regression for circuits), its unique advantages may be understated or unvalidated.

---

> ### Author Rebuttal · Authors · 2026-03-30
>
> We sincerely thank you for your positive recommendation and for recognizing the value of our design.
>
> We also note that the term “transcoder” can be ambiguous across domains. In our paper, a Cross-Layer Transcoder (CLT) refers to **a software-based replacement model used mainly in mechanistic interpretability to decompose LLM computation into sparse, interpretable cross-layer features**. Therefore, we **do not claim applicability to industrial audio/video transcoder** pipelines.
> ## I. Response to Q1: Feature Attribution Choice and Dependency
> You raised an excellent point regarding the dependency on feature attribution methods. To address this, we conducted complementary experiments validating our framework against **two other strong attribution/pruning techniques** (due to length limit, we cannot post the important table here, please kindly see **Rebuttal Table A and B** in the response to Reviewer kUtg):
>
> ACDC-Style Pruning (Perturbation-based): We found it measures feature importance well in very tight budget regimes (e.g., $K=50$) but struggles at higher budgets because CLTs are highly robust to single-feature ablations. It is also computationally heavy.
>
> Activation-Magnitude Pruning: This ranks purely by each feature’s activation magnitude toward a prompt.
>
> Our Method (FAP family): By using gradient-weighted write attribution, FAP accurately ranks medium-to-low importance features and is **highly scalable**. Additionally, our FAP-Synergy variant overcomes the limits of magnitude-only scoring by reranking boundary features based on pairwise interactions.
>
> We found that the FAP family consistently outperformed the baselines. Our final advice is that **FAP-Synergy is ideal for low-budget extraction** where outlining structural coordination is critical, while base **FAP is preferable for high-budget regimes** due to its lower computational overhead.
> ## II. Response to Q2: Scalability to Large Models
> While industrial hardware control transcoders are outside our scope, scalability to massive LLMs is indeed the primary motivation of our PIE framework. CLTs have massive feature spaces (hundreds of thousands of features), making automatic interpretation extremely computationally expensive. PIE saves significant computational resources by pruning this massive search space first. As detailed in Appendix E, by interpreting only a pruned subset ($K=100$), **PIE reduces the downstream evaluation burden by approximately 119x for Llama-3.2-1B and 106x for Gemma-2-2B** compared to a global dictionary sweep, saving substantial compute time and API costs.
>
> ## III. Response to Q3: Quantifying Interpretability
>
> We fully agree that interpretability should be evaluated quantitatively rather than only qualitatively. In fact, PIE is explicitly designed around a dual-objective quantitative evaluation protocol that measures both behavioral fidelity and explanation quality.
>
> **Behavioral fidelity.** We use two complementary metrics:
> - **KL Divergence**, which measures distribution-level fidelity by comparing the final next-token probability distribution of the original subject model and the pruned replacement model;
> - **Prediction Change Rate (PCR)**, which measures decision-level fidelity by tracking how often pruning changes the model’s argmax prediction.
>
> **Explanation quality.** We evaluate generated feature descriptions using FADE-style metrics:
> - **Clarity**: whether an auditor LLM can generate synthetic activating examples from the description alone;
> - **Purity**: whether the description distinguishes genuinely high-activating real examples from low-activating ones;
> - **Responsiveness**: whether naturally matching examples consistently trigger the feature.
>
> Importantly, these explanation metrics are evaluated on a held-out Wikipedia distribution rather than on the IOI or Docstring prompts themselves, so they assess whether the explanations capture **general feature semantics rather than only task-specific utility**. These quantitative results are **documented in Table 1 and 3, and Figure 6**.
> ## IV. Response to Q4: Hardware Deployment
> We believe this question arises from a terminology mismatch. In our paper, a Cross-Layer Transcoder (CLT) is not a physical deployment transcoder or hardware circuit. Accordingly, our evaluation **focuses on behavioral fidelity within the LLM setting**, measured by KL Divergence and PCR, rather than on FPGA/microcontroller deployment constraints such as power, thermals, or fault tolerance. We will clarify this terminology more explicitly in the revision to avoid confusion.
>
> We deeply appreciate your opinions, as it helped us strengthen the core validation of our paper. We will update our manuscript to include these new baseline comparisons and user guidelines to solve this question.
>
> If these updates and clarifications address your concerns, we would be immensely grateful if you might graciously consider updating your assessment.

---

> > ### Author Rebuttal · Reviewer_RH64 · 2026-04-03
> >
> > The response of Q3 does not change my overall assessment and I maintain my score.

---

> > > ### Author Response · Authors · 2026-04-06
> > >
> > > Thank you for taking the time to read our rebuttal and for maintaining your positive assessment.
> > >
> > > Wishing you the best with your research!

---

### Official Review · Reviewer_w8BK · 2026-03-11

**Soundness:** 3
**Presentation:** 3
**Significance:** 3
**Originality:** 2
**Overall Recommendation:** 5
**Confidence:** 2

**Summary:**

The paper introduces "Prune, Interpret, Evaluate" (PIE): a framework for Large Language Model mechanistic interpretability based on a novel pruning strategy of Cross-Layer Transcoder features (CLTs) to drastically reduce the cost of downstream feature interpretation. The paper motivates the proposed Feature Attribution Patching (FAP) pruning method by considering a number of novel efficiency metrics other than more standard evaluations. The experimental section demonstrates that the proposed pruning strategy can reduces the interpretation framework calls by a substantial factor, making mechanistic interpretability overall more accessible.

**Compliance With Llm Reviewing Policy:**

Affirmed.

**Final Justification:**

The authors addressed my concerns regarding the impact of the proposed method variant by providing convincing relative comparisons that make the value of the proposed method more clear.

**Key Questions For Authors:**

1. How is the KL measure defined in section 3.4 as a function of different sequences and tokens?
2. Can the authors provide more intuition regarding the Efficiency Metrics defined in section 5.2 and the different model aspect they capture
3. Is the benefit of FAP-synergy significant according to any of the quantitative metrics? How do the the hyper-parameter (e.g. percentage of $\mathcal{F}_{core}$ and $\mathcal{F}_{bound}$, $\lambda$) affect the metrics? Can the author provide any intuition
4. Can the author provide an example of a failure case arising from over-pruning? Making the users more aware of the risk of the framework would provide great value.

**Limitations:**

yes

**Strengths And Weaknesses:**

# Strenghts
* The paper clearly motivates the need for cheaper and more effective LLM mechanistic interpretation, providing a practical and effective solution
* Overall, the paper is well written and mostly clear. Most of the concepts are clear even if I am not an expert of the field
* The paper makes some tangible progress in a very critical area of the literature

# Weaknesses
1. While overall the paper feels self-contained, section 3.1 seems quite compressed. Additional intuition and background on CLTs inner workings may help the reader to better understand the pruning metric and efficiency scores later defined in the paper. In particular I struggled to understand how the point-wise CLT reconstruction translates into a measure of distribution divergence (section 3.4)
2. The high level intuition for the metrics introduced in section 5.2 is clear. However, I find the section lacks some intuition regarding the specific goal of each metric with respect to each other and the specific choice of functional forms.
3. The paper reports an illustrative example to motivate the use of FAP-synergy. However the quantitative metrics do not seem to support a substantial difference between the methods. The hyper-parameter exploration for FAP-synergy seems limited.
4. The paper recognizes the limitation of "over-pruning" when it comes to mechanistic interpretability. However no failure case study is provided.

---

> ### Author Rebuttal · Authors · 2026-03-30
>
> We sincerely thank you for your encouraging review and we appreciate your recognition of our work's value. Below we clarify these points and will revise the manuscript accordingly.
> ## I. Definition of the KL Measure
> CLT Replacement: CLTs decompose and reconstruct the original Transformer's computations. A faithful CLT replacement model (the two models that we selected) produces next-token logits closely matching the original subject model.
>
> Causal Pruning: To avoid computationally prohibitive and redundant interpretation, we prune the CLT to a sparse causal core by retaining a budget of K features and patching out the spatial downstream contributions of all unselected features.
>
> Measuring Divergence: We compute the KL divergence between the original subject models' and pruned replacement models' next-token probability distributions for the same prompt, measured at the final token. Pruning removes CLT feature writes at intermediate sites; these local changes alter the final residual stream, changing the final logits. KL is measured on that final output distribution. **Low KL suggests that the K retained features effectively isolated the task-relevant circuitry.** We will add more clarification in Section 3.4.
> ## II. Intuition for the Efficiency Metrics
> The efficiency metrics ask: How much interpretable meaning do we get per unit of model breakage?
>
> Semantic Cost Efficiency (SCE): Our "Yield" metric. The numerator (Clarity + Purity) represents the **total extracted semantic value**, while the denominator (KL $\times$ PCR) represents **the compounded behavioral "tax"** we paid to get it (for how far our pruned model diverges from the original model).
>
> Signal-to-Divergence Ratio (SDR): $10 \cdot \log_{10}(\frac{\text{Responsiveness}}{\text{KL}})$. Modeled directly after acoustic signal-to-noise ratios, this provides a decibel-scale metric. It treats the feature's responsiveness as the valid "signal" and the model's behavioral degradation (KL) as the "noise floor."
>
> Risk-Adjusted Responsiveness (RAR): $\frac{\text{Responsiveness}}{\text{KL}\cdot \text{PCR}}$. This evaluates the pure responsiveness of the features but actively penalizes it by the risk of model degradation (KL $\times$ PCR).
>
> We will add these explanations to our manuscript.
> ## III. FAP-Synergy Significance and Budget Sweeps
> While the KL drop at K=100 appears small, **our new supplementary budget sweeps (K=50, 100, 200, 400, 800)** on the IOI and **Doc-String tasks** (a standard structural Python code task utilized in the original ACDC paper) clarify where FAP-Synergy shines (due to length limit, please **kindly see the two tables in our rebuttal to Reviewer kUtg**):
>
> **Low-Budget Regimes (K=50, 100): FAP-Synergy excels**. In a tight budget, competition for feature selection is fierce; pairwise synergy estimation is crucial for properly outlining coordinated circuits.
>
> High-Budget Regimes (K=400, 800): As feature importance magnitude stabilizes, the reranking rate drops (under 1% at K=400, under 0.5% at K=800).
>
> Guideline: **FAP-Synergy is ideal for low-budget extraction** where structural outlines are critical; FAP is preferable for high-budget regimes due to lower computational overhead.
> ## IV. Hyperparameter Selection Intuition ($\lambda$ and $bp$)
> **Appendix A (Figure 3; Table 4)** provides a hyperparameter sensitivity sweep. Intuitively:
>
> $\lambda$ (Synergy Weight): Dictates how strongly we weigh the synergy effect. **Too high overestimates synergy** (unjustly boosting weak single effects); too low underestimates the interaction.
>
> $bp$ (Boundary Percent): The window of features considered. For K=100 and $bp$=50%, we rerank features ranked 50 to 150. **Too large $bp$ wastes compute and adds noise**.
>
> Our sweep empirically confirms that $\lambda$=3 and $bp$=25% provide the best trade-off in our experiments.
> ## V. Failure Case Study of Over-Pruning
> Users should absolutely be aware of the risks of over-pruning. Our supplementary experiments provide a good qualitative example of what breaks with strict budgets.
>
> **While computationally cheap, a strict K drops important structural information**. For example, Feature $L4.16331$ is a conjunction detector joining two coordinated noun phrases (usually two person names). For the prompt, "Then, Arthur and Ruby had a long argument, and afterwards Ruby said to Arthur," this feature correctly activates on the "and" between Arthur and Ruby, passing crucial grammatical context.
>
> At K=100, both FAP and FAP-Synergy successfully retain this feature. However, **at K=50, the feature is dropped entirely.** Losing this grammatical anchor **weakens the structural interpretability** of the resulting circuit, demonstrating the tangible cost of over-pruning.
>
>
> We will revise the manuscript accordingly. If our modifications address your previous questions, we would be especially grateful if you would consider updating your evaluation.
>
> Thank you again for your time and guidance.

---

> > ### Author Rebuttal · Reviewer_w8BK · 2026-04-02
> >
> > I thank the authors for their extensive and detailed answers, I believe that the additional explanation on the proposed metrics and mention of failure cases definitely strengthens the submission.
> >
> > After reviewing the additional results in the response to Reviewer kUtg, I am still not entirely convinced of the effectiveness of FAP-sinergy when compared to FAP. This is mostly due to the fact that the numerical differences seem minor when compared to the impact of changing the budget.

---

> > > ### Author Response · Authors · 2026-04-03
> > >
> > > Thank you for your continued engagement and for recognizing the value of our failure case study. To address your remaining concern regarding FAP-Synergy's effectiveness, we offer a reframing based on Cost Equivalence, supported by a new supplementary experiment.
> > >
> > > ### I. The "Effective Budget" (Cost Equivalence) ###
> > > Increasing the budget (K) improves KL, but drastically inflates downstream auto-interpretation costs. To quantify FAP-Synergy's efficiency, we ran a supplementary base FAP experiment at K=75 on the IOI task:
> > >
> > > Llama-3.2-1B: Base FAP at K=75 yields a KL of 1.22. FAP-Synergy at K=50 yields a KL of 1.22.
> > >
> > > Gemma-2-2B: Base FAP at K=75 yields a KL of 0.81. FAP-Synergy at K=50 yields a KL of 0.82.
> > >
> > > This demonstrates that FAP-Synergy at K=50 functionally matches the behavioral fidelity of base FAP at K=75. Because downstream evaluation costs scale linearly per feature, Synergy effectively grants the pipeline 25 "free" features, **achieving K=75 fidelity while reducing interpretation costs by 33%**.
> > >
> > > ### II. Relative Log-Space Reductions ###
> > > KL divergence is a logarithmic measure, making absolute differences at the lower end much harder to achieve. The shift from K=50 base FAP to K=50 FAP-Synergy represents an 8.2% relative reduction in KL error for Llama-3.2-1B (1.33 to 1.22), and a 9.8% relative reduction for Gemma-2-2B (0.91 to 0.82). **Shaving off ~10% of the error distribution** without adding a single feature to the evaluation budget is a highly significant algorithmic gain.
> > >
> > > While buying a larger feature budget will always improve absolute metrics, our overarching goal with PIE is to make mechanistic interpretability economically viable. FAP-Synergy is a crucial innovation precisely because it maximizes structural fidelity when compute budgets are strictly constrained.
> > >
> > > We hope this clarifies the practical significance of FAP-Synergy. If this addresses your remaining reservations, we would be incredibly grateful if you would consider updating your assessment to help us share this framework with the community.
> > >
> > > Thank you once again for your time, guidance, and constructive feedback.

---

### Official Review · Reviewer_DFvF · 2026-03-12

**Soundness:** 3
**Presentation:** 3
**Significance:** 3
**Originality:** 3
**Overall Recommendation:** 4
**Confidence:** 1

**Summary:**

**I should note that mechanistic interpretability is not my research area, and this paper may have been assigned to me in error. I would strongly recommend the Area Chair weight my review accordingly and prioritize assessments from more qualified reviewers.**

PIE is an end-to-end CLT-native framework connecting feature pruning (via gradient-weighted attribution, FAP), automatic interpretation, and FADE-style evaluation. By pruning to K=100 features before the expensive interpretation stage, PIE achieves ~40× compression over a strong random baseline at comparable behavioral fidelity on the IOI task with Llama-3.2-1B and Gemma-2-2B. A synergy-aware variant (FAP-Synergy) additionally reranks boundary features using pairwise interaction estimates.

**Compliance With Llm Reviewing Policy:**

Affirmed.

**Key Questions For Authors:**

N/A

**Strengths And Weaknesses:**

The practical motivation is clear: auto-interpretation cost dominates interpretability workflows, and pruning first is an obvious but underexplored solution.
The 40× compression result is striking and practically significant if it generalizes.
The L0.2703 case study gives concrete mechanistic substance to the synergy argument.

---

> ### Author Rebuttal · Authors · 2026-03-30
>
> We sincerely thank you for recommending our work. We are thrilled that the practical motivation of the PIE framework, the ~40x compression rate, and the L0.2703 case study resonated with you.
>
> In your recommendation, you rightly noted that generalizability is the premise of our significance. We completely agree. To strengthen the empirical validation, we have conducted extensive supplementary experiments during this rebuttal period. Specifically, we evaluated PIE across a much broader range of feature budgets ($K \in \{50, 100, 200, 400, 800\}$) and expanded our evaluation to a completely different task structure (the Doc-String task). We also implemented two new rigorous baselines: ACDC-style pruning and Activation-Magnitude pruning.
>
> **Across these expanded settings, PIE consistently outperforms the baselines in both fidelity and efficiency**. We invite you to review the **detailed tables and analysis provided in our Response to Reviewer kUtg**, which will be fully integrated into the final manuscript.
> Thank you again for your time, your positive assessment, and for helping us make the paper stronger.

---

> > ### Author Rebuttal · Reviewer_DFvF · 2026-04-04
> >
> > Thank you for your response. I will maintain my original score.

---

> > > ### Author Response · Authors · 2026-04-06
> > >
> > > Thank you again for recognizing the value and motivation behind our framework. We sincerely appreciate your time and positive assessment.
> > >
> > > Wishing you a great rest of your week!

---

### Official Review · Reviewer_kUtg · 2026-03-13

**Soundness:** 2
**Presentation:** 3
**Significance:** 2
**Originality:** 2
**Overall Recommendation:** 4
**Confidence:** 1

**Summary:**

This paper studies a practical problem in mechanistic interpretability: CLT feature spaces are large, but automatic interpretation and evaluation are expensive. The authors propose PIE (Prune, Interpret, Evaluate), a unified budget-aware pipeline for selecting, interpreting, and evaluating features. The main method, Feature Attribution Patching (FAP), ranks features using activation differences and gradient-weighted contributions, while FAP-Synergy refines features near the pruning boundary using pairwise synergy. Results show that, under a budget of K=100, FAP matches the KL fidelity of a random baseline requiring about 4,000 features, while yielding better explanation quality.

**Compliance With Llm Reviewing Policy:**

Affirmed.

**Final Justification:**

I think the paper should be scored around 3-4. I will accordingly raise it to 4 with confidence 1 (given limited knowledge of the reviewer in this domain).

Also, I just noticed that it seems all reviewers hold confidence less than 2.

So I suggest AC jointly consider the above facts to make a final decision.

**Key Questions For Authors:**

I expect the authors to address several questions in the final version.

1. Why is the primary comparison against a random baseline, rather than including stronger automated pruning or circuit discovery baselines? The current setup demonstrates the effectiveness of FAP, but it is insufficient to support stronger methodological claims.

2. FAP-Synergy plays an important role in the paper, yet its empirical improvements appear limited. Could the authors provide more systematic ablations: for example, varying the boundary percentage, different values of \lambda, or synergy effects across different K, to clarify under which regimes synergy actually provides significant benefits? The abstract claims improvements in fidelity and trade-offs, but the main text could more clearly distinguish between conceptual and empirical contributions.

3. The main experiments only report results for K=100, which the authors also acknowledge in the limitations section. If K is extended to 200, 500, or 1000, does the interpretability–fidelity trade-off between random selection and FAP remain consistent? This question directly relates to whether PIE mainly performs well in a strict-budget regime.

**Limitations:**

The authors have discussed the limitations and potential negative societal impact of their work.

**Strengths And Weaknesses:**

**Strengths**

1. The problem formulation is valuable. Existing feature-interpretation pipelines often assume uniform sampling or broad interpretation across a large number of features. However, the paper points out that among CLT features related to a target behavior, only a small subset is truly worth allocating interpretation budget to. Viewing the question of “which features to interpret” as a core research problem is both reasonable and important. This is more practically meaningful than simply proposing another feature ranking metric.

2. The design of FAP itself is reasonable. Although it is essentially a first-order attribution-style approximation, it represents a sensible engineering trade-off for the goal of quickly filtering irrelevant features from an extremely large feature space. It preserves the patching-based motivation while avoiding the high computational cost of iterative edge search. The paper also clearly emphasizes its efficiency advantage compared to methods such as ACDC.

3. The main experimental results are reasonably convincing. The paper claims that pruning to K=100 using FAP on IOI achieves fidelity roughly equivalent to that obtained by a random baseline requiring around 4k active features, corresponding to roughly a 40× compression.

**Weakness**

1. The main technical issue is that the baselines are not strong enough. In the current experimental setup, the authors primarily compare FAP/FAP-Synergy with a random baseline, while fixing the pruning budget to K = 100. (The random baseline is a necessary sanity check) However, given that the paper itself mentions ACDC, the absence of direct comparisons with closer baselines, such as attribution-based ranking, patching-derived pruning, or other circuit selection approaches, weakens the claim of methodological superiority. The current results mainly support the conclusion that the method is much better than random, rather than clearly better than existing methods.

2. The empirical gain from FAP-Synergy appears relatively limited. Although the paper spends substantial attention on the limitations of single-feature magnitude ranking and motivates synergy-aware reranking well, the experimental results suggest that FAP-Synergy functions more as a sensible refinement than as a significant improvement. The idea is conceptually sound and supported by the case study, but its empirical impact currently seems limited, which weakens its significance as a standalone contribution.

3. The experimental scope is narrow. All major results are based on IOI, which is a classic task in mechanistic interpretability but has relatively clean structure and a specific task format. If the paper aims to position PIE as a general CLT-native framework, validating it on a single behavioral task is insufficient. At least some additional tasks with different characteristics would help demonstrate that the workflow does not overly rely on the specific structure of IOI.

4. The paper relies a bit heavily on custom efficiency and cost narratives, while the associated conclusions may not be entirely robust. The authors provide cost estimates for large-scale dictionary sweeps and use these to emphasize the savings achieved by PIE in end-to-end interpretation and evaluation. While this direction is meaningful, these numbers depend heavily on assumptions about API costs, global kept-feature statistics, and the chosen budget regime. Especially given that the main experiments fix K = 100, these cost estimates are better interpreted as practical motivation rather than strong empirical evidence.

5. The authors acknowledge two important limitations themselves. First, the main pipeline is only reported under a single feature budget K=100, lacking a more systematic scaling study. Second, the work does not train CLT itself but relies on publicly available replacement checkpoints, meaning the results partially depend on the quality and coverage of these CLT models. While the limitation section is clear, it also indicates that the current work is more accurately positioned as a pruning-and-interpretation pipeline given an existing high-quality CLT, rather than a method fully validated across broader replacement-model settings.

In the end, we need to note that the reviewer is not that familiar with this topic (given the reading and understanding, suggest a score of 3-4 if all questions/weakness are addressed), and need to discuss more with other reviewers and AC in the next step.

---

> ### Author Rebuttal · Authors · 2026-03-30
>
> Thank you for your constructive feedback. Your suggestion to expand our evaluation scope was insightful. We conducted supplementary experiments to address your concerns comprehensively.
> ## I. Stronger Baselines (Addressing Weakness 1 & Q1)
>
> We agree that stronger baselines will help prove methodological superiority. Since, to our knowledge, there are currently no existing pruning methods specifically designed for CLTs, **we developed two new baselines**:
>
> **ACDC-Style Pruning**: Standard ACDC nodes (attention heads/MLP blocks) are too coarse, retaining too many features and breaking our fixed budget. We adapted it to evaluate feature-level importance via targeted activation patching during forward passes.
>
> **Activation-Magnitude Pruning**: This method measures each feature's activation magnitude on clean prompts and prunes based purely on activation ranking.
>
> As Table A and B below shows, while ACDC-style pruning accurately measures feature importance in very tight budget regimes (K=50), **the FAP family demonstrate significantly stronger long-tail robustness at higher budgets**.
> Upon inspecting the ACDC process, we found that CLTs are highly robust to single-feature ablations (due to feature redundancy or compensatory mechanisms); thus, substituting individual features (ACDC-style pruning) often yields zero effect. We observed a relatively low number of features with non-zero ACDC effects (averaging 185.18 for Gemma-2-2B and 254.54 for Llama-3.2-1B). Consequently, ACDC-style pruning degrades to random selection past K=200. Conversely, FAP's gradient estimation consistently and accurately ranks medium-to-low importance features.
>
> **FAP is also much cheaper**: one forward/backward pass versus >4000 forward passes for ACDC per prompt.
>
> ## II. Expanded Scope and Task (Addressing Weakness 2, 3, 4 & Q2, Q3)
>
> We understand that evaluating solely on the IOI task at K=100 limits our generalizability. Therefore, we included **a broader range of budgets (K=50, 100, 200, 400, 800)** and evaluated on **the Doc-String task** (a standard structural Python code task used in the original ACDC paper).
>
> The results across both tasks consistently show FAP outperforming the baselines. More importantly, this sweep clarifies the specific value of FAP-Synergy:
>
> Low-Budget Regimes (K=50, 100): FAP-Synergy excels. In tighter budgets, competition for feature selection is fierce, and pairwise synergy estimation is crucial for properly outlining coordinated circuits.
>
> High-Budget Regimes (K=400, 800): As feature importance magnitude stabilizes, the reranking rate drops (<1% at K=400, <0.5% at K=800).
>
> This provides a clear guideline: **FAP-Synergy is recommended for low-budget extraction**, whereas FAP is preferable for high-budget regimes due to its lower computational overhead. The expanded scope also demonstrates the effectiveness of the framework across a range of budget regimes, giving users flexibility to choose K based on their own computational budgets.
> ## III. Rationale for Using Public CLTs (Addressing Weakness 5)
>
> Regarding our use of public CLT models, we note the practical realities of current CLT research. Training a CLT from scratch is computationally expensive, often exceeding LLM pre-training costs at a similar scale. **Most interpretability researchers cannot train custom CLTs and must rely on public models**. By validating PIE on two of the most widely used and accessible CLTs, we ensure our work is practical for the broader community.
> ## Rebuttal Tables: General KL Across Different Ks
>
> ### Table A: IOI Task
> | Model | Method | K=50 | K=100 | K=200 | K=400 | K=800 |
> | :--- | :--- | ---: | :---: | :---: | :---: | :---: |
> | Llama-3.2-1b | FAP | 1.33 | 1.13 | 0.85 | 0.53 | 0.22 |
> | | FAP-Synergy | 1.22 | 1.12 | 0.85 | 0.53 | 0.22 |
> | | Activation-Magnitude | 1.59 | 1.52 | 1.32 | 1.07 | 0.87 |
> | | ACDC-Style Pruning | 1.26 | 1.24 | 1.22 | 1.19 | 1.13 |
> | Gemma-2-2b | FAP | 0.91 | 0.74 | 0.52 | 0.30 | 0.15 |
> | | FAP-Synergy | 0.82 | 0.73 | 0.52 | 0.30 | 0.15 |
> | | Activation-Magnitude | 1.29 | 1.25 | 1.18 | 0.96 | 0.70 |
> | | ACDC-Style Pruning | 0.81 | 0.77 | 0.76 | 0.72 | 0.64 |
>
> ### Table B: Doc-String Task
> | Model | Method | K=50 | K=100 | K=200 | K=400 | K=800 |
> | :--- | :--- | ---: | :---: | :---: | :---: | :---: |
> | Llama-3.2-1b | FAP | 0.78 | 0.69 | 0.53 | 0.32 | 0.14 |
> | | FAP-Synergy | 0.74 | 0.68 | 0.53 | 0.32 | 0.14 |
> | | Activation-Magnitude | 0.88 | 0.88 | 0.85 | 0.77 | 0.68 |
> | | ACDC-Style Pruning | 0.78 | 0.76 | 0.76 | 0.75 | 0.74 |
> | Gemma-2-2b | FAP | 0.41 | 0.36 | 0.30 | 0.21 | 0.12 |
> | | FAP-Synergy | 0.40 | 0.36 | 0.29 | 0.21 | 0.12 |
> | | Activation-Magnitude | 0.43 | 0.42 | 0.42 | 0.40 | 0.36 |
> | | ACDC-Style Pruning | 0.40 | 0.40 | 0.39 | 0.38 | 0.36 |
>
> We will update these findings in the manuscript. If these new experiments resolve your primary reservations, we would be incredibly grateful if you might consider raising your score.
>
> Thank you again for your time and advice.

---

> > ### Author Rebuttal · Reviewer_kUtg · 2026-04-04
> >
> > Thanks for the detailed response!
> >
> > Most of my concerns have been addressed, although I still believe the paper could be significantly improved in terms of comprehensiveness and solidness.
> >
> > Currently, I think the paper should be scored 3-4. I will accordingly raise it to 4 with confidence 2.

---

> > > ### Author Response · Authors · 2026-04-06
> > >
> > > Thank you for reviewing our rebuttal and updating your score. We truly appreciate your constructive feedback throughout this process.
> > >
> > > Wishing you all the best!

---

### Decision · Program_Chairs · 2026-04-30

**Decision:**

Reject

**Comment:**

Cross-Layer Transcoder (CLT) is a replacement model architecture for studying the mechanistic interpretability of large language models (LLMs). CLT will produce a large number of features. Interpreting all the features is usually too expensive and previous studies have shown that not all the features are worth interpreting. Consequently, it is intuitive to prune the features before interpretation. This paper proposed Prune, Interpret, Evaluate (PIE) which is a CLT specific and native framework for pruning, interpreting, and evaluating the CLT features. Most of the contributions of the paper is concentrated on the CLT pruning part, where, to the best of the authors' knowledge, for the first time CLT is pruned. More concretely, the paper proposed Feature Attribution Patching (FAP), a first-order approximation method adapted for CLT feature space, for pruning the CLT features via ranking feature importance scores. In addition, because two features can have synergy, the paper also proposed FAP-synergy so that lower-score but synergistically important features can possibly be retained in pruning. The paper (including the new results from rebuttal) found that models after FAP and FAP-synergy pruning can achieve better or the same interpretability using much fewer features than models after random pruning or a couple of other baselines methods. Consequently, the cost of feature interpretation and evaluation can be substantially reduced by the proposed method.

Despite the fact that the reviewers were satisfied with most of the rebuttals from the authors and the paper got recommendation ratings from all the four reviewers, the confidence scores from the reviewers were not very high. At least two reviewers explicitly mentioned that they are not an expert in the area of mechanistic interpretability.

In my own opinion, even though the major topic of the paper is mechanistic interpretability, its major technical contribution is actually on the pruning method invention (FAP-synergy) and application (FAP) for CLT features. If we take a closer look at the pruning study, it is somewhat coarse to me. The paper should definitely describe and cite the very first first order approximation pruning method for neural networks and its later variants and improvements, such as iterative pruning, and discuss whether they are applicable to CLT feature pruning. If they are not applicable, the paper should analyze and explain why they are not applicable. For example, as the authors mentioned in the rebuttal, training CLT is very expensive, so maybe iterative pruning and fine-tuning CLT weights are also expensive. Even if this is the first time CLT is pruned, because pruning has been an active research area for a long time, the paper should have discussed and compared with more of the existing pruning methods. FAP-synergy seems to be manually orchestrated based on heuristics rather than algorithmically chosen. It is possible that more advanced pruning methods can identify the synergistic features in a more systematic way and achieve better performance. The paper failed to study the possibility of this.

Overall, I think this paper makes an important step towards efficient mechanistic interpretability of CLT. However, because it lies in the intersection of mechanistic interpretability and neural network pruning, the paper should have done a better job in connecting the existing pruning methods to the new domain. Doing so might not be very difficult, maybe some of the existing pruning methods are just completely not applicable and the paper would just have to explain why. I am looking forward to seeing this paper with a more systematic and comprehensive pruning study in the future.